# Synthesis of a New β-Galactosidase Inhibitor Displaying Pharmacological Chaperone Properties for GM1 Gangliosidosis

**DOI:** 10.3390/molecules27134008

**Published:** 2022-06-22

**Authors:** Francesca Clemente, Macarena Martínez-Bailén, Camilla Matassini, Amelia Morrone, Silvia Falliano, Anna Caciotti, Paolo Paoli, Andrea Goti, Francesca Cardona

**Affiliations:** 1Dipartimento di Chimica “Ugo Schiff” (DICUS), Università di Firenze, Via Della Lastruccia 3-13, 50019 Sesto Fiorentino, Italy; macarena.martinez@iiq.csic.es (M.M.-B.); camilla.matassini@unifi.it (C.M.); andrea.goti@unifi.it (A.G.); 2Glycosystems Laboratory, Instituto de Investigaciones Químicas (IIQ), CSIC—Universidad de Sevilla, Av. Américo Vespucio 49, 41092 Sevilla, Spain; 3Department of Neurosciences, Pharmacology and Child Health (NEUROFARBA), University of Florence, Viale Pieraccini n. 24, 50139 Firenze, Italy; amelia.morrone@unifi.it; 4Laboratory of Molecular Biology of Neurometabolic Diseases, Neuroscience Department, Meyer Children’s Hospital, Viale Pieraccini n. 24, 50139 Firenze, Italy; silvia.falliano@meyer.it (S.F.); anna.caciotti@meyer.it (A.C.); 5Dipartimento di Scienze Biomediche Sperimentali e Cliniche “Mario Serio” (DSBSC), University of Florence, Viale Morgagni 50, 50134 Florence, Italy; paolo.paoli@unifi.it

**Keywords:** iminosugars, β-galactosidase inhibitors, GM1 gangliosidosis, *GLB1*, pharmacological chaperones, nitrones, Grignard reagents, reductive amination

## Abstract

GM1 gangliosidosis is a rare lysosomal disease caused by the deficiency of the enzyme β-galactosidase (β-Gal; *GLB1*; E.C. 3.2.1.23), responsible for the hydrolysis of terminal β-galactosyl residues from GM1 ganglioside, glycoproteins, and glycosaminoglycans, such as keratan-sulfate. With the aim of identifying new pharmacological chaperones for GM1 gangliosidosis, the synthesis of five new trihydroxypiperidine iminosugars is reported in this work. The target compounds feature a pentyl alkyl chain in different positions of the piperidine ring and different absolute configurations of the alkyl chain at C-2 and the hydroxy group at C-3. The organometallic addition of a Grignard reagent onto a carbohydrate-derived nitrone in the presence or absence of a suitable Lewis Acid was exploited, providing structural diversity at C-2, followed by the ring-closure reductive amination step. An oxidation-reduction process allowed access to a different configuration at C-3. The *N*-pentyl trihydroxypiperidine iminosugar was also synthesized for the purpose of comparison. The biological evaluation of the newly synthesized compounds was performed on leucocyte extracts from healthy donors and identified two suitable β-Gal inhibitors, namely compounds **10** and **12**. Among these, compound **12** showed chaperoning properties since it enhanced β-Gal activity by 40% when tested on GM1 patients bearing the p.Ile51Asn/p.Arg201His mutations.

## 1. Introduction

GM1 gangliosidosis (MIM# 230500) is a lysosomal storage disorder (LSD) caused by the deficiency of lysosomal β-galactosidase (β-Gal; *GLB1*; E.C. 3.2.1.23), an enzyme deputed to the hydrolysis of the terminal β-galactosyl residues from GM1 ganglioside, glycoproteins, and glycosaminoglycans, such as keratan-sulfate [1]. GM1 gangliosidosis is considered a neurodegenerative disorder classified into three clinical subtypes: severe infantile (type I; OMIM #230500), late-infantile/juvenile (type II; OMIM #230600), and milder adult (type III; OMIM #230650) forms [2]. The severe infantile form (type I) is fatal by 1–2 years old and associated with severe progressive neurological symptoms that manifest during early infancy. Patients with the juvenile form (type II) manifest first symptoms at 2–3 years, show the slower progression of the disease and have a relatively higher life expectancy (late childhood or early adolescence). Type III GM1 gangliosidosis is the mildest form, with later onset of symptoms (early to mid-adolescence) and higher life expectancy. The estimated incidence of GM1 gangliosidosis is in the range of 1:100.000–200.000 live births [2,3], with some isolated communities being particularly affected (e.g., Malta, 1:3.700) [4].

At present, there is no cure available for GM1 gangliosidosis. Symptomatic treatments for some of the neurologic symptoms are available but do not significantly alter the condition’s progression [2]. Therapies relying on pharmacological chaperones (PCs) may constitute a future option for the treatment of this disease. PCs are reversible inhibitors of the enzyme used in sub-inhibitory amounts, which interact with the mutated protein, favoring its proper conformation, improving its stability in the endoplasmic reticulum, preventing its aggregation and premature degradation, thus resulting in the rescue of enzymatic activity [5,6,7,8,9]. In the past twenty years, several glycomimetic small molecules have been investigated as PCs for GM1 gangliosidosis, such as 1-deoxygalactonojirimycin (DGJ (**1**); Figure 1) [10,11]. This compound was later approved for Fabry Disease, another LSD, under the trade name of Galafold^TM^ [12].

The ability of **1** to rescue the activity of mutant β-Gal up to 7-fold in human fibroblasts from patients with GM1 gangliosidosis was demonstrated in 2001 [13]. Since then, great synthetic efforts have identified more potent PCs towards GM1 gangliosidosis (Figure 1) [11]. Several *N*-alkylated DGJ derivatives were synthesized with enhanced specificity and affinity to galactosidases, such as *N*-butyl-DGJ (NB-DGJ (**2**); Figure 1) and *N*-nonyl-DGJ (NN-DGJ (**3**); Figure 1) [13,14]. The bicyclic sp^2^-iminosugar isothiourea derivative 6*S*-NBI-DGJ (5*N*,6*S*-[*N*′-butyliminomethylidene)-6-thio-1-deoxygalactonojirimycin (**4**); Figure 1) resulted in a sixfold β-Gal activity enhancement in the fibroblasts from patients homozygous for the juvenile GM1 mutation p.Arg201Cys [9,15,16] and showed promising properties as PC in a GM1 gangliosidosis mouse model [17]. A significant contribution was also provided by the Graz group, starting with the compound coined as DLHEX-DGJ (**5**), which showed significant activity enhancement (18-fold) with chaperone-sensitive p.Arg201Cys and p.Arg201His cell lines at 500 µM [18]. In addition, several C-alkylated azasugars (*C*-pentyl-4-*epi*-isofagomine **6** and its nonyl analogous **7**, Figure 1) were found to be potent and selective inhibitors of human lysosomal β-Gal able to the half-maximal recovery of β-Gal activity. However, compound **7** also strongly inhibited lysosomal β-glucosidase, which may cause undesirable side effects [19,20]. Moreover, the “*all-cis*” trihydroxypiperidines **8** and **9** (Figure 1) were suitable inhibitors of lysosomal β-Gal and were able to increase β-Gal activity in GM1 gangliosidosis patient fibroblasts up to two–sixfold (at <100 µM concentration) [21].

Based on the observation that the configuration of the carbon atoms bearing the hydroxy groups and the position of the alkyl chain play a subtle role in the biological activity of β-Gal, the straightforward stereoselective synthesis of four novel C-2 pentyl trihydroxypiperidines **10**, **11**, **12** and **13** (Figure 1) was undertaken. The new compounds are C-2 alkylated trihydroxypiperidines with two different stereochemical patterns at the hydroxy groups and the opposite absolute configuration at C-2. The choice of a pentyl alkyl chain (instead of a longer one) was made to avoid undesirable side effects due to lysosomal β-glucosidase inhibition [22,23]. 

The target compounds were obtained starting from two common intermediates **14** and **15**, which could afford, respectively, both the trihydroxypiperidines **10** and **12** through simple *O*- and *N*-deprotection, and the “*all-cis*” trihydroxypiperidines **11** and **13**, epimeric at C-3, through an oxidation-reduction sequence after temporary nitrogen protection. The piperidine intermediates **14** and **15** could be obtained via intramolecular reductive amination (RA) [24] of hydroxylamines **16** and **17** with an *S* or *R* absolute configuration at the newly formed stereocenter, in turn, derived from Grignard reagent additions onto nitrone **18** in the presence or absence of a suitable Lewis Acid (Figure 1). Nitrone **18** was readily accessed from **19** with an 85% yield by reaction with *N*-benzyl hydroxylamine in dry CH_2_Cl_2_ [25]. Aldehyde **19** was synthesized in four steps from d-mannose on a gram scale [26]. To confirm the role of the chain position on activity, compound **21** was synthesized starting from the piperidine intermediate **20** through *N*-Alkylation. The piperidine intermediate **20** was obtained from **18** by RA [23].

In this work, the synthesis of the target compounds **10–13** and **21** is described, together with the biological evaluation of their effect on the human lysosomal β-Gal enzyme and an in vitro study on fibroblast cell lines bearing the p.Ile51Asn/p.Arg201His and the p.Arg201His/Tyr83LeufsX8 mutations from juvenile GM1 gangliosidosis patients. 

## 2. Results and Discussion

### 2.1. Chemistry: Synthesis and Structural Assignment 

The addition of pentylmagnesium bromide to nitrone **18** in dry THF at −78 °C for 3 h afforded a good yield (70%, entry 1, Table 1) of the corresponding hydroxylamines **16** and **17** in a 3.5:1 ratio in favor of the hydroxylamine **16** with the (*S*) absolute configuration at the newly formed stereocenter. The addition of the same Grignard reagent in the presence of BF_3_·Et_2_O (1.0 equiv.) resulted in a reversal of stereoselectivity, and the hydroxylamine **17** with an (*R*) absolute configuration at the newly formed stereocenter was formed with a *dr* = 5.0:1 (entry 2, Table 1). In keeping with the findings of previous experiments with different Grignard reagents, the two hydroxylamines **16** and **17,** which were readily separable by flash column chromatography, were not stable in air and spontaneously partially oxidized to the corresponding nitrones **22** and **23** (Figure 2) [22,23]. Their formation was attested by ^1^H-NMR and MS analyses immediately after purification by column chromatography; complete characterization was carried out after oxidation of the hydroxylamines **16** and **17** to nitrones **22** and **23** with the hypervalent iodine reagent IBX in dry CH_2_Cl_2_ (Figure 2) [27]. 

The hydroxylamine/nitrone mixtures were employed in the ring-closure RA step with H_2_ as a reducing agent (balloon), Pd/C as a catalyst and two equivalents of acetic acid in MeOH, affording the piperidines **14** and **15**, (*S*)- and (*R*)-configured at C-2, respectively, in 2 days and excellent yields after treatment with a strongly basic anion exchange resin (Figure 3).

Careful analysis of the ^1^H-NMR, 2D-NMR and 1D-NOESY spectra carried out on piperidines **14** and **15** confirmed the stereochemical outcome of the Grignard addition. The ^1^H-NMR signals of the azasugar portion for piperidines **14** and **15**, together with their coupling constants, are shown in Figure 2. For compound **14**, 1D-NOESY spectra did not help to elucidate the structural assignment. However, its ^1^H-NMR spectra showed small couplings constants for ^3^J_2–3_, ^3^J_3–4_ and ^3^J_4–5_ (2.0 Hz, 3.2 Hz and 5.2 Hz, respectively). This pattern is in agreement with an (*S*) absolute configuration at C-2, with the piperidine displaying a preferred ^1^C_4_ conformation (slightly distorted due to the fused dioxolane ring) in which the bulky chain lies in the equatorial position and H-3 and H-4 are in an equatorial orientation (Figure 2). In the 1D-NOESY spectra of compound **15**, strong NOE correlation peaks were observed between H-2 and H-4, H-4 and H_b_-6, and H-2 and H_b_-6, which testify to their mutual 1,3-diaxial position and allow to confirm the (*R*) configuration at C-2 (see Appendix A). Indeed, the axial orientation of H-2 derives from a preferred ^4^C_1_ conformation, which accommodates the bulky chain again in an equatorial position (Figure 2). 

Final removal of the acetonide protecting groups under acidic conditions (aqueous HCl in MeOH) followed by basic treatment afforded the target trihydroxypiperidines **10** and **12** as free amines in good yields (Figure 4).

The inversion of configuration at C-3 in compounds **11** and **13** was achieved through an oxidation–reduction sequence carried out on temporarily *N*-protected piperidines (Figure 5 and Figure 6).

The protection of piperidines **14** and **15** with the *tert*-butyloxycarbonyl group to obtain compounds **24** and **25**, followed by oxidation with Dess Martin periodinane, provided the key ketone intermediates **26** and **27** in good yields over two steps (Figure 5). Both ketones underwent sodium borohydride reduction in good yields and high selectivity, showing the same preference for an attack on the *Re* face, which produced the *all-cis* relative configuration in the resulting trihydroxypiperidines (Figure 6). The high stereoselectivity observed in the reduction of **26** and **27** to **28** and **29**, respectively, can be ascribed to a strongly favoured *axial* attack of hydride at the C-3 carbonyl in the ^1^C_4_ chair conformation in both cases. Indeed, this conformation allows a kinetically favoured approach of hydride *anti* to the vicinal C-O bond, thus enjoying TS stabilization by the low-lying energy σ* of C-O bond according to the Felkin–Anh model (Figure 3) [28,29].

Final removal of the acetonide protecting groups of **28** and **29** under acidic conditions (aqueous HCl in MeOH), followed by basic treatment afforded the target trihydroxypiperidines **11** and **13** as free amines in excellent yields (92% and 98%, Figure 6).

The occurred inversion of configuration at C-3 in compounds **11** and **13** and their preferred conformation were established on the basis of careful analysis of their ^1^H-NMR and 1D-NOESY spectra. In particular, for compound **11**, a strong NOE correlation peak between H-3 and H-5 was observed (Figure 4). Together with the high coupling constants observed for the signal of H-3, this finding confirmed an axial orientation of this proton with an *ax-ax* relationship with H-2 (J = 8.0 Hz) and a 1,3-diaxial interaction with H-5. This pattern agrees with an (*S*) absolute configuration at C-3, with the piperidine displaying a preferred ^1^C_4_ conformation in which the bulky chain lies in the equatorial position. 1D-NOESY studies for compound **13** showed strong NOE peaks correlating protons H-2, H_b_-6 and H-4 (Figure 4) (see Appendix A). Moreover, the ^1^H-NMR spectrum showed broad singlets for protons H-3 and H-5, consistent with their equatorial position. These evidences confirm the (*S*) configuration at C-3 and a ^4^C_1_ conformation in which the bulky chain is equatorial.

To investigate the role played by the position of the alkyl chain on the biological activity, the trihydroxypiperidine **21** alkylated at nitrogen with a pentyl chain was also synthesized. This aim was achieved by alkylation of piperidine **20** followed by deprotection of the acetonide group of **30** (40% yield over two steps, Figure 7).

### 2.2. Preliminary Biological Screening towards Human β-Galactosidase and β-Glucosidase

It is known that 3,4,5-trihydroxipiperidine display its biological properties in its fully deprotected form [30]. Therefore, the target compounds **10**–**13** and **21** were first evaluated as human lysosomal β-Gal inhibitors at 1 mM in human leukocyte homogenates, and as human lysosomal β-Glu (β-glucosidase) inhibitors at 1 mM in order to evaluate the selectivity of the new compounds. The results are shown in Table 2. Only compounds **10** and **12** showed a good selectivity and considerable 90% and 42% inhibitory activity towards β-Gal, with a moderate IC_50_ (400 ± 15 µM and 1.15 ± 0.1 mM, respectively). The trihydroxypiperidine **10** with the (*S*) configuration at C-2 (Table 2, Entry 1) was more active than the corresponding epimeric trihydroxypiperidine **12** with the (*R*) configuration (Table 2, Entry 2). The presence of a pentyl chain in trihydroxypiperidines **10** and **12** appears to benefit the inhibitory activity, in agreement with previous results [19,20].

Conversely, the opposite configuration at C-3 as in compounds **11** and **13** resulted in a dramatic decrease of β-Gal inhibition (down to 36% and 35%, respectively, regardless of the configuration at C-2) (Table 2, entries 3 and 4 *vs.* entries 1 and 2) in line with our previous results on alkylated azasugars with a relative “*all-cis*” configuration at the hydroxy/amine-substituted stereocenters [31]. These results confuted our expectations that inhibition of β-Gal might benefit from the “*all-cis*” configuration of the three hydroxy groups. Moreover, the “*all-cis*” configuration made compounds **11** and **13** better inhibitors of β-Glu, eroding any selectivity in inhibition of the two enzymes. *N*-Alkylation in compound **21** was also deleterious for inhibition against β-Gal (Table 2, Entry 5). None of the newly synthesized compounds showed a significant inhibition towards β-glucosidase at 1 mM (from 1% to 45%). Once again, the results highlight that the presence of a longer alkyl chain is essential for providing affinity towards β-glucosidase [22,23,31,32,33,34]. 

Since literature data show that potent PCs for β-Gal can also be found among moderate and relatively good inhibitors [21], the ability of compounds **10** and **12** to rescue the enzymatic activity of β-Gal on cell lines bearing selected mutations was tested. 

Kinetic analyses were also performed to determine the mechanism of action of **12** (see Appendix A), which showed chaperoning properties (see Section 2.3). The results indicate that it acts as a noncompetitive β-Gal inhibitor, with a *K*_i_ value of 1.4 ± 0.7 mM.

### 2.3. Pharmacological Chaperoning Activity

In this experiment, the inhibitors of β-Gal **10** and **12** were incubated for four days with fibroblasts from two juvenile GM1 gangliosidosis patients bearing p.Ile51Asn/p.Arg201His and p.Arg201His/Tyr83LeufsX8 mutations, respectively, followed by assays of cell lysates for lysosomal β-Gal activity (see Appendix A). The experiments showed that all compounds are non-toxic at high concentrations in both cell lines. Compound **12** showed a β-Gal activity rescue of 1.40-fold at 600 µM on GM1 patient fibroblasts bearing the p.Ile51Asn/p.Arg201His mutations.

The results obtained suggest that only the p.Ile51Asn/p.Arg201His mutations are responsive to our compound. Interestingly, the stronger inhibitor **10** did not show any enzyme activity rescue. 

To the best of our knowledge, compound **12** represents the first example of a non-competitive β-Gal inhibitor acting as a pharmacological chaperone on GM1 patients’ cells. A chaperoning behaviour for other noncompetitive inhibitors of lysosomal enzymes has been previously observed, in particular for gene mutations leading to Gaucher, Fabry, Pompe, and Tay–Sachs diseases [35].

## 3. Materials and Methods

### 3.1. General Experimental Procedures for the Syntheses

Commercial reagents were used as received. All reactions were carried out under magnetic stirring and monitored by TLC on 0.25 mm silica gel plates (Merck F254). Column chromatographies were carried out on Silica Gel 60 (32–63 µm) or on silica gel (230–400 mesh, Merck, Darmstadt, Germany). Yields refer to spectroscopically and analytically pure compounds unless otherwise stated. ^1^H-NMR spectra were recorded on a Varian Gemini 200 MHz, a Varian Mercury 400 MHz, or on a Varian INOVA 400 MHz instrument at 25 °C. ^13^C-NMR spectra were recorded at 50 MHz or at 100 MHz. Chemical shifts are reported relative to CDCl_3_ (^1^H: δ = 7.27 ppm, ^13^C: δ = 77.0 ppm). Integrals are in accordance with assignments; coupling constants are given in Hz. For detailed peak assignments, 2D spectra were measured (g-COSY, g-HSQC) and 1D-NOESY. Small-scale microwave-assisted syntheses were carried out in a microwave apparatus for synthesis (CEM Discover) with an open reaction vessel and external surface sensor. The following abbreviations were used to designate multiplicities: s = singlet, d = doublet, t = triplet, q = quartet, m = multiplet, br s = broad singlet, and dd = double-doublet. IR spectra were recorded with an IRAffinity-1S Shimadzu spectrophotometer. ESI-MS spectra were recorded with a Thermo Scientific LCQ fleet ion trap mass spectrometer. Elemental analyses were performed with a Thermo Finnigan FLASH EA 1112 CHN/S analyzer. Optical rotation measurements were performed on a JASCO DIP-370 polarimeter.

#### 3.1.1. Synthesis of Benzyl-2,3-*O*-(1-methylethylidene)-β-l-gulofuranoside-5-(pentyl)-5-(*N*-benzyl-hydroxylamine) (**16**) and Benzyl-2,3-*O*-(1-methylethylidene)-α-d-mannofuranoside-5-(pentyl)-5-(*N*-benzyl-hydroxylamine) (**17**)

##### Method A: General Procedure without Lewis Acid

A solution of nitrone **18** (452 mg, 1.2 mmol) in dry THF (25 mL) was stirred at −78 °C under a nitrogen atmosphere, and pentylMgBr (1.8 mL, 2.2 mmol) was slowly added. The reaction mixture was stirred for 3 h until a TLC control (PEt/EtOAc 1:1) attested to the disappearance of the starting material. A 1M NaOH solution (10 mL) and Et_2_O (10 mL) were added to the mixture at 0 °C and left stirring for 20 min. The two layers were separated, and the aqueous layer was extracted with Et_2_O (2 × 10 mL). The combined organic layers were washed with brine (2 × 30 mL), dried with Na_2_SO_4_, and concentrated under reduced pressure to give a mixture of hydroxylamines **16** and **17** (**16**:**17** ratio 3.5:1; the **16**:**17** ratio was determined by integration of ^1^H-NMR signals of the crude reaction mixtures). The crude mixture was purified by silica gel column chromatography (gradient eluent from PEt/EtOAc 13:1 to 10:1) to give **16** (337 mg, 0.74 mmol, R*_f_* = 0.35, PEt/EtOAc 10:1) and **17** (46 mg, 0.10 mmol, R*_f_* = 0.25, PEt/EtOAc 10:1) corresponding to 70% total yield. 

The secondary hydroxylamines **16** and **17** spontaneously oxidize to the corresponding nitrones **22** and **23**, so only ^1^H-NMR and MS-ESI spectra were performed immediately after their purification by column chromatography.

**16**: colourless oil. ^1^H-NMR (400 MHz, CDCl_3_) δ = 7.43–7.26 (m, 10H, Ar), 5.17 (s, 1H, H-1), 4.79 (d, J = 11.6 Hz, 1H, OC*H*_2_Ar), 4.71–4.70 (m, 1H, H-3), 4.63 (d, J = 5.6 Hz, 1H, H-2), 4.58 (d, J = 11.6 Hz, 1H, OC*H*_2_Ar), 4.43 (d, J = 9.2 Hz, 1H, H-4), 4.27 (d, J = 14.0 Hz, 1H, NC*H*_2_Ar), 4.04 (d, J = 14.0 Hz, 1H, NC*H*_2_Ar), 3.30–3.26 (m, 1H, H-5), 1.54–1.51 (m, 4H, H-1′ and H-2′) 1.45 (s, 3H, C(C*H*_3_)_2_), 1.36–1.26 (m, 7H, C(C*H*_3_)_2_), H-3′–H-4′), 0.91 (t, J = 6.2 Hz, 3H, H-5′) ppm. C_27_H_37_NO_5_: mass required *m*/*z* = 455.27; mass found—MS (ESI) *m*/*z* (%) = 478.21 (100) [M + Na]^+^, 456.15 (35) [M + H]^+^.

**17**: colourless oil. ^1^H-NMR (400 MHz, CDCl_3_) δ = 7.37–7.24 (m, 10H, Ar), 5.07 (s, 1H, H-1), 4.98 (br s, OH), 4.82–4.80 (m, 1H, H-3), 4.66 (d, J = 11.6 Hz, 1H, OC*H*_2_Ar), 4.63 (d, J = 5.6 Hz, 1H, H-2), 4.50 (d, J = 11.6 Hz, 1H, OC*H*_2_Ar), 4.33–4.31 (m, 1H, H-4), 3.91 (s, 2H, NC*H_2_*Ar), 3.31 (q, J = 5.9 Hz, 1H, H-5), 1.78–1.72 (m, 2H, H-1′), 1.59–1.52 (m, 2H, H-2′), 1.43 (s, 3H, C(C*H*_3_)_2_), 1.35–1.26 (m, 7H, H-3′–H-4′ and C(C*H*_3_)_2_), 0.93 (t, J = 6.0 Hz, 3H, H-5′) ppm. C_27_H_37_NO_5_: mass required *m*/*z* = 455.27; mass found—MS (ESI) *m*/*z* (%) = 478.19 (100) [M + Na]^+^,456.15 (31) [M + H]^+^.

##### Method B: General Procedure with Lewis Acid

To a stirred solution of nitrone **18** (368 mg, 0.96 mmol) in dry THF (20 mL) at room temperature, boron trifluoride diethyl etherate (118 µL, 0.96 mmol) as Lewis acid was added, and the resulting mixture was stirred at room temperature under nitrogen atmosphere for 15 min. The reaction mixture was cooled at −30 °C and pentylMgBr (1.5 mL, 1.73 mmol) was slowly added. The reaction mixture was stirred at −30 °C for 2 h until a TLC control (PEt/EtOAc 1:1) attested to the disappearance of the starting material. A 1 M NaOH solution (10 mL) and Et_2_O (10 mL) were added to the mixture at 0 °C and left stirring for 20 min. The two layers were separated, and the aqueous layer was extracted with Et_2_O (2 × 10 mL). The combined organic layers were washed with brine (2 × 30 mL) and dried with Na_2_SO_4_, and concentrated under reduced pressure to give a mixture of hydroxylamines **16** and **17** (**16**:**17** ratio 1:5; the **16**:**17** ratio was determined by integration of ^1^H-NMR signals of the crude reaction mixtures). The crude mixture was purified by silica gel column chromatography (gradient eluent from PEt/EtOAc 13:1 to 10:1) to give **16** (50 mg, 0.11 mmol, R*_f_* = 0.35, PEt/EtOAc 10:1) and **17** (280 mg, 0.61 mmol, R*_f_* = 0.25, PEt/EtOAc 10:1) corresponding to 75% total yield.

The secondary hydroxylamines **16** and **17** spontaneously oxidize to the corresponding nitrones **22** and **23**, so only ^1^H-NMR and MS-ESI spectra were performed immediately after their purification by column chromatography. 

#### 3.1.2. Synthesis of Benzyl-2,3-*O*-(1-methylethylidene)-β-l-gulofuranoside-5-(pentyl)-5-(phenylmethanimine oxide) (**22**)

To a stirred solution of hydroxylamine **16** (19.5 mg, 0.04 mmol) in dry CH_2_Cl_2_ (2 mL), IBX (2-Iodoxybenzoic acid contains stabilizer (45 wt. %) (40 mg, 0.06 mmol) was added, and the resulting mixture was stirred under nitrogen atmosphere at room temperature for 3 h when a TLC control (PEt/EtOAc 10:1) attested the disappearance of the starting material. A saturated solution of NaHCO_3_ (4 mL) was added, and the two layers were separated and the aqueous layer was extracted with CH_2_Cl_2_ (3 × 5 mL). The combined organic layers were washed with brine (2 × 6 mL) and concentrated after drying with Na_2_SO_4_. The residue was purified by silica gel flash column chromatography (PEt/EtOAc from 10:1) to give nitrone **22** (17.7 mg, 0.039 mmol, 98%, R*_f_* = 0.25) as a straw-yellow oil.

**22**: straw yellow oil. ${\left[\alpha \right]}_{D}^{25}$ = +44.9 (c = 1.00, CHCl_3_). ^1^H-NMR (400 MHz, CDCl_3_) δ = 8.34–8.32 (m, 2H, *H*C = N and Ar), 7.46–7.43 (m, 4H, Ar), 7.30–7.23 (m, 5H, Ar), 5.01 (s, 1H, H-1), 4.77–4.74 (m, 1H, H-3), 4.67 (dd, J = 2.6, 5.8 Hz, 1H, H-2), 4.61–4.58 (m, 2H, OC*H*_2_Ar and H-4), 4.33 (d, J = 11.6 Hz, 1H, OC*H*_2_Ar), 4.08–4.03 (m, 1H, H-5), 2.17–2.11 (m, 1H, H_a_-1′), 1.69–1.64 (m, 1H, H_b_-1′), 1.50 (s, 3H, C(C*H*_3_)_2_), 1.33–1.26 (m, 9H, C(C*H*_3_)_2_ and H-2′-H-4′), 0.88–0.86 (m, 3H, H-5′) ppm. ^13^C-NMR (50 MHz, CDCl_3_) δ = 137.2 (1C, Ar), 134.4 (1C, Ar), 130.8 (1C, Ar), 130.2–127.9 (10C, Ar and H*C* = N), 112.6 (*C*(CH_3_)_2_), 104.5 (C-1), 85.4 (C-2), 79.7 (C-3), 79.1 (C-4), 76.7 (C-5), 68.8 (O*C*H_2_Ar), 31.5, 27.7, 26.2, 25.5, 24.8, 22.6 (6C, C-1′–C-4′, and C(*C*H_3_)*_2_*), 14.2 (C-5′) ppm. MS (ESI): *m*/*z* (%) = 928.62 (100) [2M + Na]^+^, 476.16 (33) [M + Na]^+^. IR (CDCl_3_): ν = 974, 1080, 1107, 1209, 1456, 1582, 2243, 2858, 2932, 3032, 3065 cm^−1^. C_27_H_35_NO_5_ (453.58): calcd. C, 71.50; H, 7.78; N, 3.09; found C, 71.81; H, 7.53; N, 2.97.

#### 3.1.3. Synthesis of Benzyl-2,3-*O*-(1-methylethylidene)-α-d-mannofuranoside-5-(pentyl)-5-(phenylmethanimine oxide) (**23**)

To a stirred solution of hydroxylamine **17** (22.2 mg, 0.05 mmol) in dry CH_2_Cl_2_ (2 mL), IBX (2-Iodoxybenzoic acid contains stabilizer (45 wt. %) (50 mg, 0.08 mmol) was added and the resulting mixture was stirred under nitrogen atmosphere at room temperature for 3 h when a TLC control (PEt/EtOAc 10:1) attested the disappearance of the starting material. A saturated solution of NaHCO_3_ (4 mL) was added and the two layers were separated and the aqueous layer was extracted with CH_2_Cl_2_ (3 × 5 mL). The combined organic layers were washed with brine (2 × 6 mL) and concentrated after drying with Na_2_SO_4_. The residue was purified by silica gel flash column chromatography (PEt/EtOAc from 10:1.2) to give nitrone **23** (22.2 mg, 0.049 mmol, 98%, R*_f_* = 0.25) as a straw-yellow oil.

**23**: straw yellow oil. ${\left[\alpha \right]}_{D}^{25}$ = +79.8 (c = 0.80, CHCl_3_). ^1^H-NMR (400 MHz, CDCl_3_) δ = 8.27–8.25 (m, 2H, *H*C = N and Ar), 7.43–7.41 (m, 4H, Ar), 7.38–7.24 (m, 5H, Ar), 5.06 (s, 1H, H-1), 4.70–4.68 (m, 1H, H-3), 4.67 (d, J = 11.6 Hz, 1H, OC*H*_2_Ar), 4.60 (d, J = 5.6 Hz, 1H, H-2), 4.49 (d, J = 11.6 Hz, 1H, OC*H*_2_Ar), 4.47–4.46 (m, 1H, H-4), 4.18–4.13 (m, 1H, H-5), 2.15–2.11 (m, 1H, H_a_-1′), 1.84–1.78 (m, 1H, H_b_-1′), 1.47 (s, 3H, C(C*H*_3_)_2_), 1.42–1.24 (m, 6H, H-2′–H-4′), 1.21 (s, 3H, C(C*H*_3_)_2_), 0.85 (t, J = 6.0 Hz, 3H, H-5′) ppm. ^13^C-NMR (100 MHz, CDCl_3_) δ = 137.5 (1C, Ar), 136.2 (1C, Ar), 130.6 (1C, Ar), 130.5–127.9 (10C, Ar and H*C* = N), 112.6 (*C*(CH_3_)_2_), 105.5 (C-1), 85.3 (C-2), 80.1 (C-4), 79.4 (C-3), 74.7 (C-5), 69.2 (O*C*H_2_Ar), 31.7, 29.7, 26.3, 25.7, 24.9, 22.7 (6C, C-1′–C-4′ and C(*C*H_3_)*_2_*), 14.2 (C-5′) ppm. MS (ESI): *m*/*z* (%) = 928.62 (100) [2M + Na]^+^, 476.18 (20) [M + Na]^+^. IR (CDCl_3_): ν = 974, 1080, 1107, 1209, 1456, 1582, 2243, 2860, 2932, 3032, 3065 cm^−1^. C_27_H_35_NO_5_ (453.58): calcd. C, 71.50; H, 7.78; N, 3.09; found C, 71.73; H, 7.58; N, 3.00.

#### 3.1.4. Synthesis of (2*S*,3*R*,4*S*,5*R*)-3-Hydroxy-4,5-*O*-(1-methylethylidene)-2-pentylpiperidine (**14**)

To a mixture of hydroxylamine **16** (314 mg, 0.69 mmol in the presence of nitrone **22**) in MeOH (35 mL), acid acetic (80 µL, 1.38 mmol) and Pd/C (160 mg) were added under a nitrogen atmosphere. The mixture was stirred at room temperature under a hydrogen atmosphere (balloon) for two days until a control by ^1^H-NMR spectroscopy attested the presence of acetate salt of 3-hydroxy-4,5-*O*-(1-methylethylidene)-2-pentylpiperidine piperidine **14**. The mixture was filtered through Celite^®,^ and the solvent was removed under reduced pressure. The corresponding free amine was obtained by dissolving the residue in MeOH, then the strongly basic resin Ambersep 900-OH was added, and the mixture was stirred for 40 min. The resin was removed by filtration and the crude product was purified on silica gel by flash column chromatography (CH_2_Cl_2_/MeOH/NH_4_OH (6%) 15:1:0.1) to afford 151 mg (0.62 mmol, 90% R*_f_* = 0.30) of **14** as a white solid.

**14**: white solid m.p. 120–122 °C. ${\left[\alpha \right]}_{D}^{25}$ = +36.05 (c = 2.00, CHCl_3_). ^1^H-NMR (400 MHz, CD_3_OD) *δ* = 4.19 (q, J = 6.3 Hz, 1H, H-5), 4.14 (dd, J = 5.2, 3.2 Hz, 1H, H-4), 3.80 (t, J = 2.6 Hz, 1H, H-3), 3.05 (dd, J = 5.6, 13.2, Hz, 1H, H_b_-6), 2.73 (td, J = 2.0, 7.0 Hz, 1H, H-2), 2.66 (dd, J = 7.6, 13.2 Hz, 1H, H_a_-6), 1.56–1.51 (m, 1H, H_a_-1′), 1.47 (s, 3H, C(C*H*_3_)_2_), 1.43–1.33 (m, 10H, H_b_-1′, C(C*H*_3_)_2_ and H-2′–H-4′), 0.92 (t, J = 7.0 Hz, 3H, H-5′) ppm. ^13^C-NMR (50 MHz, CD_3_OD) = 109.9 (*C*(CH_3_)_2_), 77.9 (C-4), 72.1 (C-5), 68.4 (C-3), 55.4 (C-2), 47.2 (C-6), 33.2, 31.3, 28.5, 26.9, 26.3, 23.7 (6C, C-1′–C-4′ and C(*C*H_3_)_2_), 14.4 (q, C-5′) ppm. MS (ESI): *m*/*z* (%) = 244.02 (100) [M + H]^+^. IR (CD_3_OD): ν = 1153, 1171, 1219, 1246, 1287, 1379, 1462, 2247, 2301, 2641, 2859, 2931, 3341 cm^−1^. C_13_H_25_NO_3_ (243.35): calcd. C, 64.16; H, 10.36; N, 5.76; found C, 64.14; H, 10.56; N, 5.60.

#### 3.1.5. Synthesis of (2*R*,3*R*,4*S*,5*R*)-3-Hydroxy-4,5-*O*-(1-methylethylidene)-2-pentylpiperidine (**15**)

To a mixture of hydroxylamine **17** (150 mg, 0.33 mmol in the presence of nitrone **23**) in MeOH (20 mL), acid acetic (38 µL, 0.66 mmol) and Pd/C (75 mg) were added under a nitrogen atmosphere. The mixture was stirred at room temperature under a hydrogen atmosphere (balloon) for two days, until a control by ^1^H-NMR spectroscopy attested the presence of acetate salt of 3-hydroxy-4,5-*O*-(1-methylethylidene)-2-pentylpiperidine piperidine **15**. The mixture was filtered through Celite^®^ and the solvent was removed under reduced pressure. The corresponding free amine was obtained by dissolving the residue in MeOH, then the strongly basic resin Ambersep 900-OH was added, and the mixture was stirred for 40 min. The resin was removed by filtration and the crude product was purified on silica gel by flash column chromatography (CH_2_Cl_2_/MeOH/NH_4_OH (6%) 15:1:0.1) to afford 72 mg (0.30 mmol, 91%, R*_f_* = 0.30) of **15** as a colourless oil.

**15**: colourless oil. ${\left[\alpha \right]}_{D}^{25}$ = −32.3 (c = 1.00, CHCl_3_). ^1^H-NMR (400 MHz, CD_3_OD) *δ* = 4.21–4.19 (m, 1H, H-5), 3.84 (t, J = 6.2 Hz, 1H, H-4), 3.32–3.24 (m, 2H, H_a_-6 and H-3), 2.93 (d, J = 13.2 Hz, 1H, H_b_-6), 2.27–2.21 (m, 1H, H-2), 1.83–1.78 (m, 1H, H_a_-1′), 1.50 (s, 3H, C(C*H*_3_)_2_), 1.37 (s, 3H, C(C*H*_3_)_2_), 1.35–1.30 (m, 7H, H_b_-1′and H-2′–H-4′), 0.92 (t, J = 6.0 Hz, 3H, H-5′) ppm. ^13^C-NMR (100 MHz, CD_3_OD) = 110.1 (*C*(CH_3_)_2_), 81.8 (C-4), 75.8 (C-3), 75.3 (C-5), 60.2 (C-2), 46.7 (C-6), 33.3–23.6 (6C, C-1′–C-4′and C(*C*H_3_)*_2_*), 14.4 (C-5′) ppm. 1D-NOESY: Irradiation of 2-H gave a NOE at H-4 and H_b_-6, and irradiation of H-4 gave a NOE at H-2 and H_b_-6. MS (ESI): *m*/*z* (%) = 244.03 (100) [M + H]^+^. IR (CD_3_OD): ν = 1161, 1220, 1244, 1381, 2247, 2642, 2858, 2930, 3334 cm^−1^. C_13_H_25_NO_3_ (243.35): calcd. C, 64.16; H, 10.36; N, 5.76; found C, 64.24; H, 10.26; N, 5.42.

#### 3.1.6. Synthesis of (2*S*,3*R*,4*R*,5*R*)-2-Pentylpiperidine-3,4,5-triol (**10**)

A solution of **14** (30 mg, 0.12 mmol) in MeOH (4 mL) was left stirring with 12 M HCl (20 µL) at room temperature for 16 h. The crude mixture was concentrated to yield the hydrochloride salt of **14**. The corresponding free amine was obtained by dissolving the residue in MeOH, then the strongly basic resin Ambersep 900-OH was added, and the mixture was stirred for 40 min. The resin was removed by filtration, and the crude product was purified on silica gel by flash column chromatography (CH_2_Cl_2_/MeOH/NH_4_OH (6%) 2:1:0.05) to afford 22 mg (0.11 mmol, 92%, R*_f_* = 0.40) of **10** as the free base as a white solid. 

**10**: white solid m.p. 125–127 °C. ${\left[\alpha \right]}_{D}^{25}$ = +15.1 (c = 1.00, CH_3_OH). ^1^H-NMR (400 MHz, CD_3_OD) δ = 3.88–3.85 (m, 1H, H-3), 3.83 (dd, J = 2.8, 8.0 Hz, 1H, H-5), 3.69–3.68 (m, 1H, H-4), 2.80–2.79 (m, 1H, H-2), 2.75 (d, J = 8.0 Hz, 2H, H-6), 1.48–1.33 (m, 8H, H-1′–H-4′), 0.91 (t, J = 6.0 Hz, 3H, H-5′) ppm. ^13^C-NMR (100 MHz, CD_3_OD) = 72.5 (C-3), 71.6 (C-4), 67.1 (C-5), 54.1 (C-2), 46.9 (C-6), 33.1, 31.6, 27.0, 23.7 (4C, C-1′–C-4′), 14.4 (C-5′) ppm. MS (ESI): *m*/*z* (%) = 204.07 (100) [M + H]^+^. C_10_H_21_NO_3_ (203.28): calcd. C, 59.09; H, 10.41; N, 6.89; found C, 59.00; H, 10.30; N, 6.80.

#### 3.1.7. Synthesis of (2*R*,3*R*,4*R*,5*R*)-2-Pentylpiperidine-3,4,5-triol (**12**)

A solution of **15** (30 mg, 0.12 mmol) in MeOH (4 mL) was left stirring with 12 M HCl (20 µL) at room temperature for 16 h. The crude mixture was concentrated to yield the hydrochloride salt of **15**. The corresponding free amine was obtained by dissolving the residue in MeOH, then the strongly basic resin Ambersep 900-OH was added, and the mixture was stirred for 40 min. The resin was removed by filtration and the crude product was purified on silica gel by flash column chromatography (CH_2_Cl_2_/MeOH/NH_4_OH (6%) 2:1:0.05) to afford 20 mg (0.10 mmol, 83%, R*_f_* = 0.40) of **12** as the free base, as a white solid.

**12**: white solid m.p. 74–76 °C. ${\left[\alpha \right]}_{D}^{25}$ = −16.8 (c = 0.50, CH_3_OH). ^1^H-NMR (400 MHz, CD_3_OD) δ = 3.85 (br s, 1H, H-5), 3.36–3.29 (m, 2H, H-4 and H-3), 2.95 (d, J = 14.0 Hz, 1H, H_a_-6), 2.67 (d, J = 14.0 Hz, 1H, H_b_-6), 2.30–2.28 (m, 1H, H-2), 1.87–1.82 (m, 1H, H_a_-1′), 1.56–1.52 (m, 1H, H_a_-2′), 1.34–1.33 (m, 6H, H_b_-1′, H_b_-2′, H-3′–H-4′), 0.92 (t, J = 6.0 Hz, 3H, H-5′) ppm. ^13^C-NMR (50 MHz, CD_3_OD) = 76.7 (C-4), 73.9 (C-3), 70.7 (C-5), 61.7 (C-2), 50.8 (C-6), 33.4, 32.9, 26.3, 23.6 (4C, C-1′–C-4′), 14.4 (C-5′) ppm. MS (ESI): m/z (%) = 204.06 (100) [M + H]^+^. C_10_H_21_NO_3_ (203.28): calcd. C, 59.09; H, 10.41; N, 6.89; found C, 59.11; H, 10.21; N, 6.79.

#### 3.1.8. Synthesis of (2*S*,3*R*,4*S*,5*R*)-3-Hydroxy-4,5-*O*-(1-methylethylidene)-*N*-Boc-2-pentylpiperidine (**24**)

To a stirred solution of **14** (90 mg, 0.37 mmol) and NaHCO_3_ (47 mg, 0.56 mmol) in H_2_O (2.5 mL), MeOH (2.5 mL), and Boc_2_O (121 mg, 0.56 mmol) were added. The mixture was stirred at room temperature for 48 h until a TLC control (CH_2_Cl_2_/MeOH/NH_4_OH (6%) 10:1:0.1) attested to the disappearance of the starting material. The mixture was concentrated and extracted with EtOAc (3 × 20 mL). The combined organic layers were washed with brine, dried over Na_2_SO_4_ and concentrated to give **24** (117 mg, 0.34 mmol, 92%) as a white solid.

**24**: white solid m.p. 86–88 °C. ${\left[\alpha \right]}_{D}^{25}$ = +8.05 (c = 2.00, CHCl_3_). ^1^H-NMR (400 MHz, CDCl_3_) *δ* = 4.19 (br s, 1H, H-5), 4.15–4.08 (m, 3H, H-4, H-2 and H_a_-6), 3.92 (t, J = 6.0 Hz, 1H, H-3), 3.14 (d, J = 14.4 Hz, 1H, H_b_-6), 2.81 (br s, 1H, OH), 1.70–1.65 (m, 1H, H_a_-1′), 1.43 (s, 9H, C(C*H*_3_)_3_), 1.41 (s, 3H, C(C*H*_3_)_2_), 1.30 (s, 3H, C(C*H*_3_)_2_), 1.29–1.25 (m, 7H, H_b_-1′, H-2′–H-4′), 0.85 (t, J = 6.0 Hz, 3H, H-5′) ppm. ^13^C-NMR (100 MHz, CDCl_3_) *δ* = 155.9 (N*C*OO), 109.1 (*C*(CH_3_)_2_), 79.9 (*C*(CH_3_)_3_), 76.2 (C-4), 73.2 (C-5), 69.1 (C-3), 53.6 (C-2), 40.2, 39.7 (C-6, two rotamers), 28.1 (3C, C(*C*H_3_)_3_), 31.8, 27.6, 26.1, 25.6, 25.5, 22.7 (6C, C-1′–C-4′ and C(*C*H_3_)*_2_*), 14.1 (C-5′) ppm. MS (ESI): *m*/*z* (%) = 708.84 (100) [2M + Na]^+^, 366.09 (56) [M + Na]^+^. IR (CDCl_3_): ν = 950, 1124, 1454, 1685, 2922, 3604 cm^−1^. C_18_H_33_NO_5_ (343.46): calcd. C, 62.95; H, 9.68; N, 4.08; found C, 62.80; H, 10.00; N, 4.03.

#### 3.1.9. Synthesis of (2*R*,3*R*,4*S*,5*R*)-3-Hydroxy-4,5-*O*-(1-methylethylidene)-*N*-Boc-2-pentylpiperidine (**25**)

To a stirred solution of **15** (190 mg, 0.78 mmol) and NaHCO_3_ (98 mg, 1.17 mmol) in H_2_O (3 mL), MeOH (3 mL), and Boc_2_O (252 mg, 1.17 mmol) were added. The mixture was stirred at room temperature for 48 h until a TLC control (CH_2_Cl_2_/MeOH/NH_4_OH (6%) 10:1:0.1) attested to the disappearance of the starting material. The mixture was concentrated and extracted with EtOAc (3 × 20 mL). The combined organic layers were washed with brine, dried over Na_2_SO_4_, and the crude product was purified on silica gel by flash column chromatography (CH_2_Cl_2_/MeOH/NH_4_OH (6%) 15:1:0.1) to afford 237 mg (0.69 mmol, 88%, R*_f_* = 0.34) of **25** as a white solid.

**25**: white solid m.p. 73–75 °C. ${\left[\alpha \right]}_{D}^{25}$ = −19.1 (c = 0.55, CHCl_3_). ^1^H-NMR (400 MHz, CDCl_3_) *δ* = 4.38–4.28 (m, 1H, H-5), 4.02–3.94 (m, 2H, H-4 and H_a_-6), 3.78–3.68 (m, 2H, H-2 and H-3), 2.91 (br s, 1H, OH), 2.68 (br s, 1H, H_b_-6), 1.66 (br s, 2H, H-1′), 1.45 (s, 3H, C(C*H*_3_)_2_), 1.41 (s, 9H, C(C*H*_3_)_3_), 1.31 (s, 3H, C(C*H*_3_)_2_), 1.26–1.22 (m, 6H, H-2′–H-4′), 0.84–0.83 (m, 3H, H-5′) ppm. ^13^C-NMR (50 MHz, CDCl_3_) *δ* = 155.4 (N*C*OO), 109.9 (*C*(CH_3_)_2_), 80.2 (*C*(CH_3_)_3_), 78.3 (C-4), 72.3 (C-3), 70.8 (C-5), 56.4, 55.8 (C-2 two rotamers), 41.3, 40.6 (C-6, two rotamers), 28.4 (3C, C(*C*H_3_)_3_), 31.9, 31.2, 29.8, 27.6, 25.2, 22.6 (6C, C-1′–C-4′ and C(*C*H_3_)*_2_*), 14.0 (C-5′) ppm. MS (ESI): *m*/*z* (%) = 366.09 (100) [M + Na]^+^, 708.84 (39) [2M + Na]^+^. IR (CDCl_3_): ν = 947, 1161, 1456, 1684, 2928, 3595 cm^−1^. C_18_H_33_NO_5_ (343.46): calcd. C, 62.95; H, 9.68; N, 4.08; found C, 62.88; H, 9.55; N, 4.00.

#### 3.1.10. Synthesis of (2*S*,4*S*,5*R*)-3-Oxo-4,5-*O*-(1-methylethylidene)-*N*-Boc-2-pentylpiperidine (**26**)

Dess-Martin periodinane (216 mg, 0.51 mmol) was added to a solution of **24** (116 mg, 0.34 mmol) in dry CH_2_Cl_2_ (11 mL) at room temperature. The reaction mixture was stirred for 3 h until a TLC control (CH_2_Cl_2_/MeOH/NH_4_OH (6%) 15:1:0.1) attested the disappearance of the starting material. The mixture was extracted with NaHCO_3_ saturated solution, dried over Na_2_SO_4_ and concentrated. The residue was purified by silica gel flash chromatography (Hex/EtOAc 5:1) to give **26** (93 mg, 0.27 mmol, 79%, R*_f_* = 0.41) as a white solid.

**26**: white solid m.p. 104–106 °C. ${\left[\alpha \right]}_{D}^{25}$ = +32.1 (c = 1.00, CHCl_3_). ^1^H-NMR (400 MHz, CDCl_3_) *δ* = 4.75–4.64 (m, 1H, H-2), 4.49–4.47 (m, 1H, H-5), 4.39 (br s, 1H, H-4), 4.29 (d, J = 6.4 Hz, 1H, H_a_-6), 2.95 (br s, 1H, H_b_-6), 1.78–1.71 (m, 1H, H_a_-1′), 1.59–1.51 (m, 1H, H_b_-1′), 1.44 (s, 9H, C(C*H*_3_)_3_), 1.41 (s, 3H, C(C*H*_3_)_2_), 1.30 (s, 3H, C(C*H*_3_)_2_), 1.27–1.25 (m, 6H, H-2′–H-4′), 0.83 (t, J = 6.0 Hz, 3H, H-5′) ppm. ^13^C-NMR (100 MHz, CDCl_3_) *δ* = 205.6 (*C* = O), 155.2 (N*C*OO), 111.6 (*C*(CH_3_)_2_), 80.6 (*C*(CH_3_)_3_), 76.7 (C-4 or C-5), 76.6 (C-4 or C-5), 61.7, 60.6 (C-2, two rotamers), 42.8, 41.5 (C-6, two rotamers), 28.4 (3C, C(*C*H_3_)_3_), 31.6, 29.3, 26.5, 25.2, 25.0, 22.6 (6C, C-1′–C-4′ and C(*C*H_3_)*_2_* ), 14.0 (q, C-5′) ppm. MS (ESI): *m*/*z* (%) = 364.00 (100) [M + Na]^+^. IR (CDCl_3_): ν = 1159, 1217, 1256, 1370, 1410, 1456, 1692, 1744, 2927 cm^−1^. C_18_H_31_NO_5_ (341.45): calcd. C, 63.32; H, 9.15; N, 4.10; found C, 63.00; H, 9.25; N, 3.99.

#### 3.1.11. Synthesis of (2*R*,4*S*,5*R*)-3-Oxo-4,5-*O*-(1-methylethylidene)-*N*-Boc-2-pentylpiperidine (**27**)

Dess-Martin periodinane (218 mg, 0.75 mmol) was added to a solution of **25** (171 mg, 0.50 mmol) in dry CH_2_Cl_2_ (15 mL) at room temperature. The reaction mixture was stirred for 3 h until a TLC control (CH_2_Cl_2_/MeOH/NH_4_OH (6%) 15:1:0.1) attested the disappearance of the starting material. The mixture was extracted with NaHCO_3_ saturated solution, dried over Na_2_SO_4_ and concentrated. The residue was purified by silica gel flash chromatography (Hex/EtOAc 4:1) to give **27** (137 mg, 0.40 mmol, 80%, R*_f_* = 0.23) as a colourless oil.

**27**: colourless oil. ${\left[\alpha \right]}_{D}^{25}$ = +41.0 (c = 0.30, CHCl_3_). ^1^H-NMR (400 MHz, CDCl_3_) *δ* = 4.88–4.79 (m, 1H, H-5), 4.63–4.53 (m, 2H, H-4 and H_a_-6), 4.46–4.32 (m, 1H, H-2), 2.66–2.64 (m, 1H, H_b_-6), 1.99 (br s, 1H, H_a_-1′), 1.61–1.4 (m, 13H, H_b_-1′, C(C*H*_3_)_3_ and C(C*H*_3_)_2_), 1.38 (s, 3H, C(C*H*_3_)_2_), 1.32–1.26 (m, 6H, H-2′–H-4′), 0.89–0.88 (m, 3H, H-5′) ppm. ^13^C-NMR (100 MHz, CDCl_3_) *δ* = 206.6 (*C* = O), 154.8 (N*C*OO), 112.3 (*C*(CH_3_)_2_), 81.3 (*C*(CH_3_)_3_), 79.2 (C-4), 74.3 (C-5), 63.0, 62.2 (C-2, two rotamers), 45.0, 43.5 (C-6, two rotamers), 28.4 (3C, C(*C*H_3_)_3_), 31.8, 28.8, 27.2, 25.7, 25.5, 22.7 (6C, C-1′–C-4′ and C(*C*H_3_)*_2_*), 14.1 (q, C-5′) ppm. MS (ESI): *m*/*z* (%) = 364.08 (100) [M + Na]^+^. IR (CDCl_3_): ν = 1159, 1217, 1256, 1369, 1410, 1456, 1692, 1744, 2928 cm^−1^. C_18_H_31_NO_5_ (341.45): calcd. C, 63.32; H, 9.15; N, 4.10; found C, 63.13; H, 9.10; N, 3.97.

#### 3.1.12. Synthesis of (2*S*,3*S*,4*S*,5*R*)-3-Hydroxy-4,5-*O*-(1-methylethylidene)-*N*-Boc-2-pentylpiperidine (**28**)

A solution of **26** (60 mg, 0.18 mmol) in EtOH (1.4 mL) was cooled to 0 °C and NaBH_4_ (17 mg, 0.45 mmol) was added. The reaction mixture was allowed to warm to room temperature and stirred for 18 h until a TLC control (Hex/EtOAc 4:1) attested to the disappearance of the starting material. Then, water (0.3 mL) and MeOH (0.8 mL) were added, and the mixture was stirred for 10 h at room temperature and concentrated under reduced pressure. The crude product was purified by silica gel flash chromatography (CH_2_Cl_2_/MeOH 10:1) to give **28** (45 mg, 0.13 mmol, 72%, R*_f_* = 0.50) as a white solid.

**28**: white solid m.p. 104–106 °C. ${\left[\alpha \right]}_{D}^{25}$ = +6.5 (c = 1.00, CHCl_3_). ^1^H-NMR (400 MHz, CDCl_3_) *δ* = 4.43–4.40 (m, 1H, H-5), 4.27 (br s, 1H, H-3), 4.16–4.09 (m, 1H, H_a_-6), 3.92–3.78 (m, 1H, H-2), 3.57 (t, J = 7.4 Hz, 1H, H-4), 2.74–2.70 (m, 1H, H_b_-6), 2.40–2.33 (m, 1H, OH), 1.72–1.67 (m, 1H, H_a_-1′), 1.58–1.50 (m, 1H, H_b_-1′), 1.43 (s, 9H, C(C*H*_3_)_3_), 1.40 (s, 3H, C(C*H*_3_)_2_), 1.32 (s, 3H, C(C*H*_3_)_2_), 1.37–1.26 (m, 6H, H-2′–H-4′), 0.84 (t, J = 6.6 Hz, 3H, H-5′) ppm. ^13^C-NMR (100 MHz, CDCl_3_) *δ* = 155.8 (N*C*OO), 109.5 (*C*(CH_3_)_2_), 79.8 (*C*(CH_3_)_3_), 74.5 (C-5), 73.6 (C-3), 69.3 (C-4), 54.5, 53.6 (C-2 two rotamers), 42.7, 41.6 (C-6, two rotamers), 28.5 (3C, C(*C*H_3_)_3_), 32.2–26.2, 25.5, 24.7, 24.5, 22.7 (6C, C-1′–C-4′and C(*C*H_3_)*_2_*), 14.1 (C-5′) ppm. MS (ESI): *m*/*z* (%) = 366.09 (100) [M + Na]^+^, 708.83 (30) [2M + Na]^+^. IR (CDCl_3_): ν = 955, 1157, 1416, 1684, 2918, 3620 cm^−1^. C_18_H_33_NO_5_ (343.46): calcd. C, 62.95; H, 9.68; N, 4.08; found C, 62.88; H, 9.55; N, 4.00.

#### 3.1.13. Synthesis of (2*R*,3*S*,4*S*,5*R*)-3-Hydroxy-4,5-*O*-(1-methylethylidene)-*N*-Boc-2-pentylpiperidine (**29**)

A solution of **27** (70 mg, 0.21 mmol) in EtOH (1.4 mL) was cooled to 0 °C and NaBH_4_ (20 mg, 0.53 mmol) was added. The reaction mixture was allowed to warm to room temperature and stirred for 18 h until a TLC control (Hex/EtOAc 4:1) attested to the disappearance of the starting material. Then, water (0.3 mL) and MeOH (0.8 mL) were added, and the mixture was stirred for 10 h at room temperature and concentrated under reduced pressure. The crude product was purified by silica gel flash chromatography (CH_2_Cl_2_/MeOH 10:1) to give **29** (55 mg, 0.16 mmol, 76%, R*_f_* = 0.50) as a white solid.

**29**: white solid m.p. 71–73 °C. ${\left[\alpha \right]}_{D}^{25}$ = −20.3 (c = 0.65, CHCl_3_). ^1^H-NMR (400 MHz, CDCl_3_) *δ* = 4.27–4.13 (m, 2H, H_a_-6 and H-5), 4.14–4.12 (m, 1H, H-4), 4.06 (br s, 1H, H-3), 3.86–3.81 (m, 1H, H-2), 3.08–2.97 (m, 1H, H_b_-6), 2.45 (br s, 1H, OH), 1.77–1.75 (m, 1H, H-1′), 1.51 (s, 3H, C(C*H*_3_)_2_), 1.43 (s, 9H, C(C*H*_3_)_3_), 1.35 (s, 3H, C(C*H*_3_)_2_), 1.28–1.23 (m, 6H, H-2′–H-4′), 0.86–0.87 (m, 3H, H-5′) ppm. ^13^C-NMR (100 MHz, CDCl_3_) *δ* = 154.9 (N*C*OO), 109.5 (*C*(CH_3_)_2_), 80.1 (*C*(CH_3_)_3_), 75.4 (C-4), 71.6 (C-5), 66.5 (C-3), 53.9 (C-2), 41.3, 40.2 (C-6, two rotamers), 28.0 (3C, C(*C*H_3_)_3_), 31.9, 28.0, 27.0, 25.6, 24.8, 22.8 (6C, C-1′–C-4′and C(*C*H_3_)*_2_*), 14.1 (C-5′) ppm. MS (ESI): *m*/*z* (%) = 366.00 (100) [M + Na]^+^, 708.84 (40) [2M + Na]^+^. IR (CDCl_3_): ν = 954, 1157, 1416, 1684, 2918, 3620 cm^−1^. C_18_H_33_NO_5_ (343.46): calcd. C, 62.95; H, 9.68; N, 4.08; found C, 62.90; H, 9.65; N, 4.16.

#### 3.1.14. Synthesis of (2*S*,3*S*,4*R*,5*R*)-2-Pentylpiperidine-3,4,5-triol (**11**)

A solution of **28** (45 mg, 0.13 mmol) in MeOH (6 mL) was left stirring with 12 M HCl (30 µL) at room temperature for 16 h. The crude mixture was concentrated to yield the hydrochloride salt of **28**. The corresponding free amine was obtained by dissolving the residue in MeOH, then the strongly basic resin Ambersep 900-OH was added, and the mixture was stirred for 40 min. The resin was removed by filtration to afford 24 mg (0.12 mmol, 92%) of **11** as the free base as a white solid. 

**11**: white solid m.p. 130–132 °C. ${\left[\alpha \right]}_{D}^{25}$ = + 41.4 (c = 0.80, CH_3_OH). ^1^H-NMR (400 MHz, CD_3_OD) δ = 3.95 (br s, 1H, H-4), 3.56 (t, J = 6.6 Hz, 1H, H-5), 3.12 (d, J = 8.0 Hz, 1H, H-3), 2.74 (d, J = 7.6 Hz, 2H, H-6), 2.64 (t, J = 8.8 Hz, 1H, H-2), 1.83–1.77 (1H, H_a_-1′), 1.51–1.49 (m, 1H, H_a_-2′), 1.34–1.22 (m, 6H, H_b_-1′, H_b_-2′ and H-3′–H-4′), 0.92 (t, J = 6.6 Hz, 3H, H-5′) ppm. ^13^C-NMR (100 MHz, CD_3_OD) = 74.2 (C-3), 73.2 (C-4), 70.3 (C-5), 55.4 (C-2), 46.5 (C-6), 33.4, 32.8, 26.4, 23.7 (4C, C-1′–C-4′), 14.4 (C-5′) ppm. 1D-NOESY: Irradiation of H-5 gave a NOE at H-3, and irradiation of H-3 gave a NOE at H-5. MS (ESI): *m*/*z* (%) = 226.10 (100) [M + H]^+^. C_10_H_21_NO_3_ (203.28): calcd. C, 59.09; H, 10.41; N, 6.89; found C, 59.12; H, 10.23; N, 6.51.

#### 3.1.15. Synthesis of (2*R*,3*S*,4*R*,5*R*)-2-Pentylpiperidine-3,4,5-triol (**13**)

A solution of **29** (24 mg, 0.07 mmol) in MeOH (3 mL) was left stirring with 12 M HCl (15 µL) at room temperature for 16 h. The crude mixture was concentrated to yield the hydrochloride salt of **29**. The corresponding free amine was obtained by dissolving the residue in MeOH, then the strongly basic resin Ambersep 900-OH was added, and the mixture was stirred for 40 min. The resin was removed by filtration to afford 14 mg (0.069 mmol, 98%) of **13** as free base, as a pale-yellow oil. 

**13**: pale yellow oil. ${\left[\alpha \right]}_{D}^{25}$ = −40.4 (c = 0.50, CH_3_OH). ^1^H-NMR (400 MHz, CD_3_OD) δ = 3.79 (br s, 1H, H-5), 3.73 (br s, 1H, H-3), 3.46 (br s, 1H, H-4), 3.02 (d, J = 15.0 Hz, 1H, H_a_-6), 2.69 (d, J = 14.0 Hz, 1H, H_b_-6), 2.44 (t, J = 6.8 Hz, 1H, H-2), 1.58–1.53 (m, 1H, H_a_-1′), 1.51–1.44 (m, 1H, H_a_-2′), 1.43–1.29 (m, 6H, H_b_-1′, H_b_-2′, H-3′–H-4′), 0.92 (t, J = 6.0 Hz, 3H, H-5′) ppm. ^13^C-NMR (100 MHz, CD_3_OD) = 72.6 (C-3), 71.3 (C-4), 71.7 (C-5), 60.0 (C-2), 51.6 (C-6), 33.2, 32.7, 26.7, 23.7 (4C, C-1′–C-4′), 14.4 (C-5′) ppm. 1D-NOESY: Irradiation of H-2 gave a NOE at H-4 and H_b_-6, and irradiation of H-4 gave a NOE at H-2 and H_b_-6. MS (ESI): m/z (%) = 204.12 (100) [M + H]^+^. C_10_H_21_NO_3_ (203.28): calcd. C, 59.09; H, 10.41; N, 6.89; found C, 59.17; H, 10.11; N, 6.70.

#### 3.1.16. Synthesis of (3*R*,4*S*,5*R*)-1-Pentyl-5-hydroxy-3,4-*O*-(1-methylethylidene)-piperidine (**30**)

To a solution of compound **20** (96 mg, 0.55 mmol) in acetonitrile (2 mL) and Milli-Q water (0.6 mL), pentylbromide (87 µL, 0.83 mmol) and potassium carbonate (115 mg, 0.83 mmol) were added. The mixture was stirred under microwave irradiation at 120 °C for 3 h, until a TLC control attested to the disappearance of the starting material (CH_2_Cl_2_/MeOH/NH_4_OH (6%) 15:1:0.1). The mixture was filtered through Celite^®,^ and the solvent was removed under reduced pressure. The crude reaction was purified by silica gel flash column chromatography (eluent CH_2_Cl_2_/MeOH/NH_4_OH (6%) 15:1:0.1) obtaining compound **30** (62 mg, 0.25 mmol, 45%, R*_f_* = 0.34) as a waxy solid.

**30:** waxy solid. ${\left[\alpha \right]}_{D}^{24}$ = +15.0 (c = 0.50, CHCl_3_). ^1^H-NMR (400 MHz, CDCl_3_) δ = 4.27 (q, J = 6.0 Hz, 1H, H-3), 4.03 (t, J = 4.6 Hz, 1H, H-4), 3.93–3.92 (m, 1H, H-5), 2.87 (br s, 1H, OH), 2.72 (dd, J = 6.0, 12.0 Hz, 1H, H_a_-2), 2.55 (d, J = 12.0 Hz, 1H, H_a_-6), 2.46 (dd, J = 5.2, 11.6 Hz, 1H, H_b_-6), 2.40–2.33 (m, 3H, H_b_-2 and H-1′), 1.49 (s, 3H, C(C*H*_3_)_2_), 1.47–1.41 (m, 2H, H-2′), 1.34 (s, 3H, C(C*H*_3_)_2_), 1.31–1.20 (m, 4H, H-3′–H-4′), 0.87 (t, J = 7.0 Hz, 1H, H-5′) ppm. ^13^C-NMR (100 MHz, CDCl_3_) δ = 109.4 (*C*(CH_3_)_2_), 77.0 (C-4), 72.2 (C-3), 67.7 (C-5), 57.9 (C-1′), 56.0 (C-2 or C-6), 55.6 (C-2 or C-6), 29.7, 28.4, 26.6, 26.5, 22.7 (5C, C-2′–C-4′and C(*C*H_3_)*_2_*), 14.1 (C-5′) ppm. MS (ESI): *m*/*z* (%) = 244.20 (100) [M + H]^+^. IR (CDCl_3_): ν = 1059, 1381, 1468, 2247, 2826, 2862, 2935, 2989, 3478 cm^−1^. C_13_H_25_NO_3_ (243.35): calcd. C, 64.16; H, 10.36; N, 5.76; found C, 64.34; H, 10.11; N, 5.70.

#### 3.1.17. Synthesis of (3*R*,5*R*)-1-Pentyl-3,4,5-trihydroxy-piperidine (**21**) 

A solution of **30** (42 mg, 0.17 mmol) in MeOH (5 mL) was left stirring with 12 M HCl (25 µL) at room temperature for 16 h. The crude mixture was concentrated to yield the hydrochloride salt of **21**. The corresponding free amine was obtained by dissolving the residue in MeOH, then the strongly basic resin Ambersep 900-OH was added, and the mixture was stirred for 40 min. The resin was removed by filtration to afford 30 mg (0.15 mmol, 88%) of **21** [36] as the free base, as a white solid.

**21**: white solid m.p. 63–66 °C. ${\left[\alpha \right]}_{D}^{24}$ = −39.7 (c = 0.61, CH_3_OH). ^1^H-NMR (400 MHz, CD_3_OD) δ = 3.91–3.90 (m, 1H, H-3), 3.83–3.78 (m, 1H, H-5), 3.42 (br s, 1H, H-4), 2.84–2.81 (m, 2H, H_a_-2 and H_a_-6), 2.43- 2.36 (m, 2H, H-1′), 2.34–2.30 (m, 1H, H_b_-6), 2.14 (br s, 1H, H_b_-2), 1.56–1.51 (m, 2H, H-2′), 1.39–1.26 (m, 4H, H-3′–H-4′), 0.92 (t, J = 6.0 Hz, 3H, H-5′) ppm. ^13^C-NMR (100 MHz, CD_3_OD) = 75.1 (C-4), 69.4 (C-5), 69.0 (C-3), 59.3 (C-1′), 58.1 (C-2 or C-6), 57.4 (C-2 or C-6), 30.8, 27.1, 23.6, (3C, C-2′–C-4′), 14.4 (C-5′) ppm. MS (ESI): *m*/*z* (%) = 204.08 (100) [M + H]^+^. C_10_H_21_NO_3_ (203.15): calcd. C, 59.09; H, 10.41; N, 6.89; found C, 59.22; H, 10.33; N, 6.81.

### 3.2. Biological Screening towards Human Lysosomal β-Galactosidase (β-Gal) and β-Glucosidase (GCase)

All experiments on biological materials were performed in accordance with the ethical standards of the institutional research committee and with the 1964 Helsinki Declaration and its later amendments. In keeping with ethical guidelines, all blood and cell samples were obtained for storage and analysed only after written informed consent of the patients (and/or their family members) was obtained, using a form approved by the local Ethics Committee (Codice Protocollo: Lysolate “Late onset Lysosomal Storage Disorders (LSDs) in the differential diagnosis of neurodegenerative diseases: development of new diagnostic procedures and focus on potential pharmacological chaperones (PCs). Project ID code: 16774_bio, 5 May 2020, Comitato Etico Regionale per la Sperimentazione Clinica della Regione Toscana, Area Vasta Centro, Florence, Italy).

Controls and patients’ samples were anonymized and used only for research purposes.

The new compounds were screened at 1 mM concentration towards β-Galactosidase (β-Gal) and β-Glucosidase (GCase) in leukocytes isolated from healthy donors (controls). 

Isolated leukocytes were disrupted by sonication, and a Micro BCA Protein Assay Kit (Sigma–Aldrich, St. Louis, MO, USA) was used to determine the total protein amount for the enzymatic assay, according to the manufacturer’s instructions.

#### 3.2.1. Human Lysosomal β-Galactosidase (β-Gal) Activity

β-Gal activity was measured in a flat-bottomed 96-well plate. Compounds **10**–**13** and **21** solution (3 μL), 4.29 μg/μL leukocytes homogenate 1:10 (7 μL), and substrate 4-methylumbelliferyl β-d-galactopyranoside (1.47 mM, 20 μL, Sigma–Aldrich) in acetate buffer (0.1 M, pH 4.3) containing NaCl (0.1 M) and sodium azide (0.02%) were incubated at 37 °C for 1 h. The reaction was stopped by the addition of sodium carbonate (200 μL; 0.5 M, pH 10.7) containing Triton X-100 (0.0025%), and the fluorescence 4-methylumbelliferone released by β-galactosidase activity was measured in SpectraMax M2 microplate reader (λex = 365 nm, λem = 435 nm; Molecular Devices). Inhibition is given with respect to the control (without compound). Percentage β-Gal inhibition is given with respect to the control (without compound). Data are mean SD (n = 3). For compounds showing β-Gal inhibitory activity higher than 40% at 1 mM concentration, the IC_50_ values were determined by measuring the initial hydrolysis rate with 4-methylumbelliferyl β-d-galactopyranoside (1.47 mM). Data obtained were fitted by using the appropriate equation (for more details, see the Appendix A).

#### 3.2.2. Human Lysosomal β-Glucosidase (GCase) Activity

GCase activity was measured in a flat-bottomed 96-well plate. Compound solution (3 µL), 4.29 µg/µL leukocytes homogenate (7 µL), and substrate 4-methylumbelliferyl-β-d-glucoside (3.33 mM, 20 µL, Sigma–Aldrich) in citrate/phosphate buffer (0.1:0.2, M/M, pH 5.8) containing sodium taurocholate (0.3%) and Triton X-100 (0.15%) at 37 °C were incubated for 1 h. The reaction was stopped by the addition of sodium carbonate (200 µL; 0.5 M, pH 10.7) containing Triton X-100 (0.0025%), and the fluorescence of 4-methylumbelliferone released by β-glucosidase activity was measured in SpectraMax M2 microplate reader (λex = 365 nm, λem = 435 nm; Molecular Devices). A percentage GCase inhibition is given with respect to the control (without compound). Data are mean SD (n = 3).

### 3.3. Kinetic Analysis for Compound 12 vs. β-Gal

The action mechanism of compound **12** was determined studying the dependence of the main kinetic parameters (Km and Vmax) on the inhibitor concentration. Kinetic data were analysed using the Lineweaver-Burk plot (for more details, see Appendix A).

### 3.4. Pharmacological Chaperoning Activity

Fibroblasts with the p.Ile51Asn/p.Arg201His and the p.Arg201His/Tyr83LeufsX8 mutations from Juvenile GM1 patients were obtained from Meyer Children’s Hospital (50139 Firenze, Italy). 

Fibroblast cells (15.0 × 10^4^) were seeded in T25 flasks with DMEM supplemented with fetal bovine serum (10%), penicillin/streptomycin (1%), and glutamine (1%) and incubated at 37 °C with 5% CO_2_ for 24 h. The medium was removed, and fresh medium containing the compounds was added to the cells and incubated for four days. The medium was removed, and the cells were washed with PBS and detached with trypsin to obtain cell pellets, which were washed four times with PBS, frozen, and lysed by sonication in water. Enzyme activity was measured as reported above. Reported data are mean S.D. (n = 2).

## 4. Conclusions

Four new trihydroxypiperidines with a C-2 pentyl chain with both configurations were synthesized, together with their “*all-cis*” hydroxy epimers, with the aim of finding new human lysosomal β-Gal inhibitors and potential PCs for GM1-gangliosidosis.

The synthesis exploited the addition of pentylmagnesium bromide to nitrone **18**, derived from aldehyde **19**, in the presence or absence of Lewis acid followed by RA, to yield the two 2-pentyl 3,4,5-trihydroxypiperidines. The inversion of configuration at C-3 was achieved through an oxidation–reduction sequence. To investigate the role of the chain position on the activity, compound **21** was synthesized for comparison, starting from the piperidine intermediate **20** via *N*-alkylation. 

Biologic tests of the new compounds showed that **10** and **12** are β-Gal inhibitors with a moderate IC_50_ (400 ± 15 µM and 1.15 ± 0.1 mM). Kinetic analyses revealed a noncompetitive mode of inhibition for **12**, which also showed chaperoning properties, with *K*_i_ value of 1.4 ± 0.7 mM. Moreover, good selectivity towards β-Gal with respect to β-Glu was observed. The poor β-Glu inhibitory activity of all compounds confirmed that the presence of a longer linear alkyl chain (at least eight carbon atoms) is essential to impart strong β-Glu inhibitory activity.

Testing compounds **10** and **12** as potential PCs in fibroblasts from juvenile GM1 gangliosidosis patients bearing the p.Ile51Asn/p.Arg201His and the p.Arg201His/Tyr83LeufsX8 mutations highlighted that only **12** allows an activity rescue of β-Gal (40% at 600 µM) on GM1 patients bearing the p.Ile51Asn/p.Arg201His mutations, thus representing, to the best of our knowledge, a unique example of a non-competitive inhibitor with chaperoning ability for GM1 gangliosidosis. The here reported fibroblasts, which derive from juvenile GM1 gangliosidosis patients, share the p.Arg201His mutation in one allele. Thus, the rescue of the in vitro system bearing the bi-allelic composition p.Ile51Asn/Arg201His can be ascribed to the chaperone activity of compound **12** on the p.Ile51Asn mutation-bearing allele. 

It has been previously reported that the p.Ile51Asn mutation, replacing a nonpolar residue in a hydrophobic pocket into a polar residue, is likely to adversely affect the fold of the β-Gal protein [3]. This biochemical characteristic makes the p.Ile51Asn mutation particularly prone to an enzymatic stabilisation and a rescue of β-Gal activity induced by a chaperone, as demonstrated by the use of compound **12**. 

## Data Availability

The authors confirm that the data supporting the findings of this study are available within the article and/or its Appendix A.

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
