# Peer review of "Synthesis of a New β-Galactosidase Inhibitor Displaying Pharmacological Chaperone Properties for GM1 Gangliosidosis"

_molecules, 2022, doi:10.3390/molecules27134008_

Round 1

Reviewer 1 Report

The authors (F. Clemente, et al.) described the synthesis of new candidates for β-galactosidase (β-Gal) inhibitor and their biological evaluations.  Although there are few notable methods in the synthetic route, all products were obtained with good to excellent yields, and those assignments by NMR would also support their configurations.  In addition, it is interesting results that all-cis configuration candidates only show inhibition towards β-Gal, and the IC50 value of compound 10 would have been better than I expected.

Although the introduction would be a bit complicated, this manuscript and SI were well organized and I think this paper would be acceptable for Molecule after addressing the points below.

1.        The arrows in Figure 1 should be shown by retrosynthetic double arrows such as ‘=>’ which is easy to understand.

Author Response

Reviewer 1

The authors (F. Clemente, et al.) described the synthesis of new candidates for β-galactosidase (β-Gal) inhibitor and their biological evaluations.  Although there are few notable methods in the synthetic route, all products were obtained with good to excellent yields, and those assignments by NMR would also support their configurations.  In addition, it is interesting results that all-cis configuration candidates only show inhibition towards β-Gal, and the IC50 value of compound 10 would have been better than I expected. Although the introduction would be a bit complicated, this manuscript and SI were well organized and I think this paper would be acceptable for Molecule after addressing the points below.

Thank you for appreciating the synthetic part of the manuscript. We have tried to reduce the complexity of the Introduction by slightly shortening it, as also requested by Reviewer 2.

  1. The arrows in Figure 1 should be shown by retrosynthetic double arrows such as ‘=>’ which is easy to understand.

Thank you for noting. Done.

Reviewer 2 Report

Clemente et al. investigated an interesting research titledSynthesis of a new β-galactosidase inhibitor displaying pharmacological chaperone properties for GM1 gangliosidosis”. The research is scientifically sound and the manuscript is well written, and it can be accepted for publication in Molecules journal after the minor modifications listed below.

1.      Remove the words "I," "we," and "our" from the manuscript.

2.      The research objective and significance should be expressed more explicitly in the abstract. The results and conclusion in the abstract, extend the sentences.

3.      The introduction is too long; cut the information and make it more concise wherever possible.

4.      Section 2.2, why were only compounds 10-13 and 21 chosen to be evaluated for biological properties, rather than others? Critical reasons for the observed outcomes should be provided by the author in the manuscript.

Author Response

Reviewer 2

Clemente et al. investigated an interesting research titled “Synthesis of a new β-galactosidase inhibitor displaying pharmacological chaperone properties for GM1 gangliosidosis”. The research is scientifically sound and the manuscript is well written, and it can be accepted for publication in Molecules journal after the minor modifications listed below.

 We thank the Reviewer for appreciating the work.

  1. Remove the words "I," "we," and "our" from the manuscript.

Thank, this has been done. All sentences are now in the passive form.

  1. The research objective and significance should be expressed more explicitly in the abstract. The results and conclusion in the abstract, extend the sentences.

The Abstract has been thoroughly rewritten and we deem that now it is clearer.

  1. The introduction is too long; cut the information and make it more concise wherever possible.

Thank you for this suggestion. The Introduction was shortened by removing some details that were too medical and not highly significant in this context. We deem that in this revised version the Introduction is more clear and smoother.

  1. Section 2.2, why were only compounds 10-13 and 21 chosen to be evaluated for biological properties, rather than others? Critical reasons for the observed outcomes should be provided by the author in the manuscript.

Thank you for asking. Indeed, it is well known that 3,4,5-trihydroxypiperidines and their N- and O- derivatives display their biological properties in their fully deprotected form [Simone et al. Chem Biol Drug Des 2018, 92, 1171-1197, DOI: 10.1111/cbdd.13182]. For this reason, we limited the biological evaluation to the target compounds 10-13 and 21. For clarity, we added the word “target compounds” in Section 2.2 and the above-mentioned review article. The following references were renumbered consequently.